# Long-Term Results of Below-The-Knee Bypass Using a Prosthetic Graft with a Distal Arteriovenous Fistula Interposition

**DOI:** 10.3390/diagnostics13071246

**Published:** 2023-03-26

**Authors:** Francesco Spinelli, Giuseppe Roscitano, David Barillà, Graziana Derone, Antonio Nenna, Nunzio Montelione, Vincenzo Catanese, Andrea Cutrupi, Martina Maria Giambra, Alessandra Varrà, Pier Francesco Veroux, Francesco Stilo

**Affiliations:** 1Division of Vascular Surgery, Department of Medicine and Surgery, University Hospital Foundation Campus Bio-Medico, 00128 Rome, Italy; 2Department of General Surgery and Medical Specialties, University of Catania, 95100 Catania, Italy; 3Vascular and Endovascular Surgery Unit, Grande Ospedale Metropolitano “Bianchi-Melacrino-Morelli”, 89124 Reggio Calabria, Italy; 4Vascular Surgery Department, Ospedale dell’Angelo, 30171 Venezia Mestre, Italy; 5Cardiovascular Surgery Unit, Department of Medicine and Surgery, University Hospital Foundation Campus Bio-Medico, 00128 Rome, Italy

**Keywords:** chronic limb-threatening ischemia, surgical revascularization, distal adjunct, arteriovenous fistula, prosthetic graft

## Abstract

Surgical bypass is the gold standard treatment in patients affected by chronic limb-threatening ischemia in advanced GLASS stages, according to the Global Vascular Guidelines. For patients in whom an autologous graft is not available, a prosthesis could be used with the adjunct of a distal arteriovenous fistula interposition. The aim of this study was to examine the long-term results of below-the-knee surgical revascularization using a prosthesis with the distal adjunct mentioned above. From 2010 to 2020, we performed 159 lower limb below-the-knee surgical revascularizations using a prosthesis with the creation of an arteriovenous fistula interposition on the distal anastomosis. The GLASS stage was 3 in 100% of patients. The primary patency rates were as follows: 86.7% at 1 year, 57.2% at 3 years, and 12.6% at 5 years. The graft thrombosis rates were 17.4% at 1 year, 42.1% at 3 years, and 64.5% at 5 years. The amputation-free survival rates were 79% at 1 year, 76% at 3 years, and 64% at 5 years. PTFE prosthetic bypass for below-the-knee arteries using an arteriovenous fistula interposition is a good solution in patients without an autologous conduit. This technique offers reasonable graft patency and limb salvage rates.

## 1. Introduction

According to the Global Vascular Guidelines published in 2019 [1], surgical bypass grafting for lower limb revascularization is the gold standard treatment for patients affected by WIfI 3–4 and GLASS 3 chronic limb-threatening ischemia (CLTI) stages. A good-quality great saphenous vein (GSV) is the optimal autologous conduit for infrainguinal bypass surgery. Alternative (small saphenous vein or arm vein) or spliced veins are acceptable bypass conduits, although there is a higher frequency of reinterventions, and durability is inferior to single-segment GSV grafts. Prosthetic conduits may be useful in selected patients lacking other revascularization options. The surgical adjunct of a distal Ascer arteriovenous fistula (AVF) interposition may also improve the patency of prosthetic bypasses to tibial targets. In patients lacking a suitable GSV, the individual surgeon’s experience may dictate practice. The aim of our study was to analyze the patency, limb salvage, and survival rates of patients affected by WIfI 3–4 and GLASS 3 CLTI in patients submitted to below-the-knee surgical revascularization using a prosthetic graft with a distal Ascer arteriovenous fistula interposition.

## 2. Materials and Methods

A retrospective multicenter study was conducted including all consecutive patients who underwent below-the-knee surgical revascularization using a prosthetic graft with a distal Ascer arteriovenous fistula interposition for CLTI between January 2010 and December 2020 at the Vascular Surgery Units of two high-volume centers for distal revascularization in Italy. CLTI was defined as peripheral artery occlusive disease with rest pain (Rutherford category 4) and/or trophic disorder, ulcer or gangrene minor tissue loss, nonhealing ulcer, or focal gangrene (Rutherford category 5); major tissue loss extending above the transmetatarsal level (Rutherford category 6) [2]; or GLOBAL classification WIfI 3-4/GLASS 3 stages [1].

Below-the-knee surgical revascularization using a prosthetic graft with a distal Ascer AVF interposition was defined as an infrainguinal heparin-bonded PTFE bypass graft (Propaten; WL Gore & Associates, Inc., Flagstaff, AZ, USA) to below-the-knee arteries (anterior/posterior tibial artery or peroneal artery) with suitable comitans veins to perform an AVF. Patients who had undergone previous endovascular/surgical revascularization were included. Patients undergoing below-the-knee bypasses for indications other than CLTI (acute limb ischemia, Buerger’s disease or other vasculitis, thrombosed popliteal aneurysm, or popliteal entrapment) were excluded from this study. Patients who underwent femoropopliteal artery bypass grafts were also excluded. Patients were deemed unsuitable for below-the-knee prosthetic bypass graft with a distal Ascer AVF interposition if they were permanently bedridden or had one of the following clinical/anatomical criteria: unsuitable comitans vein, deep vein thrombosis, or New York Heart Association (NYHA) heart failure class III–IV.

This is a retrospective study, and according to Italian law, patient informed consent was not necessary due to the retrospective nature of the study. All patient data were collected in an anonymous database from which personal information cannot be drawn.

The planning of the procedure was based on thorough duplex ultrasound arterial mapping. Computed tomography (CT) angiography or intraoperative angiography were performed only in cases of technical difficulties that hindered [3] adequate ultrasound imaging. Target artery and comitans vein diameters were measured in B-mode on a transverse plane using ultrasound scanning with the patient in a supine position.

The inflow artery was the last patent vessel with regular morphological status and a triphasic waveform upon Doppler examination. The inflow arteries used included the external iliac artery and the common and superficial femoral arteries. The outflow artery was any patent tibial or peroneal vessel segment, even if not in continuity with the inframalleolar arteries. Comitans vein and artery characteristics were evaluated with duplex ultrasound imaging with a 7 MHz linear probe at a 60° insonation angle.

General anesthesia, epidural, nerve blockade, local anesthesia, or a combination of these were used. All surgeons in our units are trained to perform extreme bypass surgery.

Patients affected by WIfI 3–4 and GLASS 3 CLTI stages and with inadequate venous material were proposed to undergo below-the-knee prosthetic bypass graft with a distal Ascer AVF interposition.

Surgical cut-downs were performed depending on the sites of the inflow and outflow arteries. The PTFE graft was tunneled through the subcutaneous passage created once the arteries and veins were surgically dissected. To obtain good results, it is important to prepare all the vessels with careful dissection, avoiding bleeding, especially of the comitans veins, which need extensive mobilization in order to make the arteriovenous fistula without tension.

Patients received anticoagulation before clamping (50 UI/kg).

First, an AVF is usually performed between the comitans vein and the target artery without putting a clamp on the distal part of the artery in order to avoid trauma to the tiny heavily calcified vessel. This 7/0 monofilament continuous suture anastomosis was performed in a termino-lateral fashion, transecting the comitans vein just distally to the arteriotomy. Backbleeding was reduced by putting the patient in an extreme Trendelenburg position, with the head raised to avoid discomfort. Distal occlusion was then achieved by gentle external compression with a finger [4,5,6]. When the AVF was performed, proximal 5/0 monofilament continuous suture anastomosis was created between the PTFE graft and the inflow artery in a termino-lateral fashion. Then, the distal anastomosis between the PTFE prosthesis and the vein of the AVF was performed in a termino-lateral fashion (Figure 1).

Intraoperative duplex ultrasound was performed to evaluate the adequacy of the bypass and to evaluate the need to reduce AVF flow in order to avoid a heart overload by measuring the popliteal vein gradient pressure. A gradient pressure higher than 30 mm Hg was an indicator to reduce the popliteal vein blood flow by performing banding with a PTFE strip or a silk suture (Figure 2).

Single antiplatelet and anticoagulant medications were given to all patients unless dual antiplatelet therapy was indicated for concomitant cardiovascular disease or prosthetic graft. Follow-up with duplex ultrasound was performed at 1, 3, and 6 months and every 6 months thereafter.

CT angiography was sometimes performed to show the bypass course (Figure 3).

Surgical debridement or minor amputations were performed in patients with necrotic lesions as needed in the same operation or in the following days, and they were followed up with ambulatory wound care weekly.

### 2.1. Outcomes

Primary outcomes included primary patency, limb salvage, and overall survival rate. Secondary outcomes included amputation-free survival, assisted primary patency, and functional status [7]. Outcomes were defined according to Rutherford reporting standards [2]. Demographic variables and cardiovascular risk factors, such as smoking, hypertension, diabetes, and renal insufficiency, were recorded.

### 2.2. Statistical Analysis

Patient demographics, comorbidities, clinical and anatomical characteristics, procedure-related data, complications, and outcomes were collected in a dedicated Excel (Microsoft Inc., Redmond, WA, USA) database. Categorical variables were reported as absolute numbers and percentages. Continuous variables were reported as mean ± standard deviation or median. Primary, assisted, and secondary patency were estimated using Kaplan–Meier survival curves. Statistical analysis was performed with Stata (StataCorp LLC, College Station, TX, USA).

## 3. Results

Between January 2010 and December 2020, 2563 patients underwent lower limb revascularization procedures at our centers. A total of 159 patients (16%) matched the inclusion criteria for below-the-knee surgical revascularization using a prosthetic graft with a distal Ascer AVF interposition. All limbs matched the criteria for WIfI 3–4 and GLASS 3 CLTI stages.

One hundred and eight patients had already undergone previous endovascular revascularization procedures. These involved treatment of the inflow artery in 23 cases (common and external iliac arteries). In the remaining cases, 25 patients underwent endovascular procedures for femoropopliteal lesions and 60 patients for infrapopliteal lesions.

Fifty-eight patients had previously undergone failed below-the-knee surgical revascularization using the great saphenous vein, sometimes combined with the small saphenous vein. One hundred and one patients did not meet the criteria for surgical revascularization using the autologous veins for the following reasons: 35 patients had undergone previous saphenectomy for chronic venous insufficiency, and the remaining 66 patients did not have a GSV of adequate diameter and length and the small saphenous veins were also not suitable.

Risk factors and comorbidities are reported in Table 1.

The mean age was 73.8 ± 7.4 years old, and 70% were smokers. Patients were affected by diabetes (68%), hypertension (62%), chronic obstructive pulmonary disease (51%), coronary artery disease (50%), chronic kidney disease with a creatinine value more than 1.3 mg/dL (31%), and end-stage renal disease in hemodialysis (13%). The WIfI stage was 3 in 89 patients (56%) and 4 in 70 patients (44%), and the GLASS stage was 3 in 100% of patients. In Table 2, the operation details are reported.

In 27 patients (17%), the inflow was the external iliac artery, in 124 patients (78%), the inflow was the common femoral artery, and in 8 patients (5%), it was the superficial femoral artery. The run-off artery was the peroneal artery (71 patients, 45%), the anterior tibial artery (45 patients, 28%), or the posterior tibial artery (43 patients, 27%). The PTFE graft was passed in a subcutaneous tunnel along the medial side of the leg between the surgical cut-downs of the run-in and run-off arteries, but in 35 patients, the PTFE graft was passed in an unusual subcutaneous tunnel along the anterior and lateral side of the leg. This kind of tunnel was made when access to the anterior tibial artery or peroneal artery was made in the anterior or lateral region of the leg, respectively. In detail, when the peroneal artery was accessed by the lateral side, segmentation of the fibula was necessary.

In seven patients in whom the distal target vessel was the peroneal artery, by the medial access of the leg, we used an interposition of the posterior tibial vein due to thrombosis or inadequate diameter of the peroneal comitans vein.

In 40% of cases, we had to perform vein banding to reduce the gradient below 30 mmHg.

Antiplatelet therapy was administered to 153 patients (96%), and 73 (46%) of these received double antiplatelet therapy. Six (4%) patients with concomitant atrial fibrillation continued antagonist vitamin K therapy alone. Seventeen patients simultaneously received antagonist vitamin K therapy and antiplatelet therapy.

Fourteen patients (8.8%) were lost to follow-up. A total of 144 patients (91%) underwent regular follow-up until graft thrombosis, major amputation, or death. One patient did not undergo follow-up because he died on the first postoperative day from acute myocardial infarction.

Thirty-six patients experienced surgical site infections with subsequent wound dehiscence. Sixteen patients required revision with surgical wound reconstruction (eight dehiscences were in the distal anastomosis site). Eighteen patients (four dehiscences were in the distal anastomosis site) underwent longer healing because vacuum therapy was applied to the surgical wound after revision, with coverage of the artery and prosthesis. Two patients underwent PTFE removal because of infection with the creation of a pseudoaneurysm in the proximal anastomosis site.

Primary patency rates were as follows: 86.7% at 1 year, 57.2% at 3 years, and 12.6% at 5 years. During follow-up, 40 patients underwent endovascular maintenance of the bypass because the run-in or run-off worsened. Five of these 40 patients underwent percutaneous transluminal angioplasty (PTA) and stenting of the iliac arteries using covered stents. The other 35 underwent PTA of the distal anastomosis (15, 9.4%) or of the outflow vessel (20, 13%). Since 2015, we have used drug-eluting balloons to perform these endovascular interventions (Table 3).

Figure 4 shows the final angiographic result after PTA.

Assisted primary patency rates were 86.3% at 1 year, 57.4% at 3 years, and 18.6% at 5 years. Graft thrombosis in all cases determined the loss of secondary patency, so the rates are the same. In detail, the graft thrombosis rates were 17.4% at 1 year, 42.1% at 3 years, and 64.5% at 5 years. Despite this, the amputation rates were 10.4% at 1 year, 31.7% at three years, and 49.2% at 5 years. There were 48 (30%) major limb amputations during the follow-up period. During the follow-up period, 89 patients (56%) died. The amputation-free survival rates were 79% at 1 year, 76% at 3 years, and 64% at 5 years (results are reported in Figure 5 and Figure 6).

## 4. Discussion

In recent years, endovascular procedures have improved in number and outcomes. Open surgical peripheral revascularization is limited to extreme lower limb ischemia stages, as reported in the recent Global Vascular Guidelines [1]. Autogenous vein graft material has better outcomes compared to a prosthesis [1,8,9], with the prevalence of in situ GSV usage [9,10,11]. When the ipsilateral GSV is not available or not adequate, the small saphenous vein, cephalic vein, and basilic vein are feasible alternatives with acceptable patency rates [12].

The lack of a suitable great saphenous vein is the main cause of peripheral surgical revascularization failure. When an adequate GVS is unavailable and no other autologous grafts are available for surgical revascularization, the use of a prosthetic graft is mandatory to perform a below-the-knee bypass graft. One common cause of arterial bypass failure is exceedingly high outflow resistance, particularly when prosthetic grafts are placed in the infrapopliteal arteries [13]. Adjunctive procedures to enhance prosthetic graft patency include vein patches and cuffs and the creation of dAVFs. In 1979, Siegman [14] first described the use of a vein cuff that would reduce the discrepancy and permit a gradual transition of flow dynamics between graft and artery. The Miller collar was developed in 1984 [15] and is formed from a segment of vein sutured to the circumference of the arteriotomy, but it differs from it in that the graft is sutured at a 45-degree angle to the cuff. Because of this, there is a slightly wider opening at the anastomosis, and blood flow is enhanced through the graft. The Taylor patch [16] is constructed by first making a short U-shaped slit in the toe segment of the graft and then suturing the heel of the graft to the cephalad end of the arteriotomy. An ellipse-shaped vein patch is then sutured to the graft and onto the distally extended arteriotomy. Tyrrell and Wolf [17] incorporated the advantages of both the collar and patch by creating a boot-shaped venous reservoir to which the prosthetic graft is anastomosed. These methods focus on the vein as a compliance enhancer and anastomotic facilitator. Dean and Read were the first to report the beneficial effects of an AVF on a prosthetic graft in terms of improved patency in a canine model [18]. The beneficial effects of an AVF on a prosthetic graft in terms of improved patency in vivo were reported by Ascer et al. [19]. In our series, we report the results of 159 patients who matched the inclusion criteria for below-the-knee surgical revascularization using a prosthetic graft with a distal Ascer AVF interposition.

Our patients were characterized by the usual pathologies that affect most of the population that is heading toward CLTI.

We prefer the creation of a distal AVF because the interposed vein acts not only as a fistula for venous decompression but also serves as a vein cuff for improved compliance. The creation of an arteriovenous fistula helps decrease the mismatch between the prosthesis and the below-the-knee vessels, which have, especially in patients with heavily calcified arteries, very small diameters and high resistance outflows that can cause rapid thrombosis. It has been demonstrated in animal models that the interposition of vein tissue at the distal anastomosis creates a biologic “buffer zone” that leads to a decrease in hyperplasia development [20,21].

We prefer to use a 6 mm diameter PTFE graft because we think that less mismatch between the graft and the target vessels could improve patency. Kreinberg et al. [22] reported results of bypass with a distal vein cuff using a 6 mm PTFE, and they explained that it is the recipient artery that determines the flow rates of these grafts.

However, bypass grafting to below-the-knee vessels is very challenging and is performed only in a few centers with high expertise in this field, compared to the endovascular approach, which is more widely available. Arteriography use was very limited for diagnostic purposes, whereas the planning of bypass procedures was based on accurate duplex ultrasound arterial mapping. Although heavily calcified arteries may be very challenging to study only with ultrasound imaging, artery patency can be evaluated with hemodynamic information provided by a precise duplex imaging study [3].

In our experience, in the below-the-knee district, duplex ultrasonography can be more sensitive than arteriography in detecting a patent target vessel with a very slow flow. Moreover, it is harmless and less expensive than arteriography. However, test quality depends on the operator’s ability.

All patients submitted to a below-the-knee surgical revascularization using a prosthetic graft with a distal Ascer AVF interposition were patients with rest pain (Rutherford category 4) and/or trophic disorder, ulcer or gangrene minor tissue loss, nonhealing ulcer, or focal gangrene (Rutherford category 5); major tissue loss extending above the transmetatarsal level (Rutherford category 6) [2]; or GLOBAL classification WIfI 1-2/GLASS 1-2 stages with previous failed endovascular/surgical revascularization and without a suitable GSV. Primary and assisted primary patency and limb salvage are not the same as an autologous distal bypass graft, but this intervention represents the last option for revascularization in this kind of patient before a major amputation.

Our results, as shown by Kaplan–Meier curves, demonstrate good patency rates. At 1, 3, and 5 years, the primary patency rates were 86.7%, 57.2%, and 12.6%, respectively. These results are similar to those demonstrated by Almasri et al. [23]. In their systematic review, they included more than 8000 patients who underwent infrainguinal revascularization. In patients revascularized using the great saphenous vein, primary and secondary patencies at 1 and 2 years were 87%/78% and 94%/87%, respectively. The difference between these and our results is more evident in secondary/assisted patency rates; our results demonstrate an acceptable and similar primary patency rate to the vein graft and a lower assisted patency rate than the vein graft. Despite this, the bypass with the distal adjunct allows the limb to be saved, avoiding major amputation and restoring and improving Rutherford class. In some cases, loss of patency due to graft thrombosis did not mean the loss of the leg because of the creation of collateral circles; healing of the extremity wounds occurred as long as the bypass worked.

In 2015, Dardik et al. [24] published the results of 502 crural bypasses with arteriovenous fistulas obtained in 34 years. They divided this long duration into four periods and found progressive improvement in primary and secondary patency going forward. In the last period, the primary patency rates at 1 and 3 years rose to 70% and 46%. They used several graft conduits, and, unlike our experience, the technique they used differed in the arteriovenous fistula and distal anastomosis diameters. It may be that the reduced size of these anastomoses is another factor that helped reduce the turbulence that could cause graft thrombosis.

As demonstrated in the literature, patients with successful infrainguinal procedures have better results than patients submitted to major amputations [25,26]. We reported 84% limb salvage at 2 years, with only 10 major amputations, and 85% survival. Moreover, the majority of patients had good functional recovery within the first 6 months after surgery. These results encourage us to consider the below-the-knee bypass as a viable approach, even in patients with moderate-to-high surgical risk profiles.

Patency results at 5 years were not satisfactory, and they were worse than in the first 2 years. This is caused by the known progression of atherosclerotic lesions; as reported in the literature, compared to autologous vein graft, prostheses have poor results in long-term patency also in bypasses without distal adjuncts in arteries of larger diameter [1].

This study has some limitations. It is a retrospective study, and as such it has an inherent risk of bias. The number of patients included is relatively low, and therefore the sensitivity of subgroup and statistical analyses was reduced. Moreover, as referral centers, some of the patients included had previous interventions at other centers, and therefore the technical details about previous interventions were not always available.

## 5. Conclusions

PTFE prosthetic bypass to below-the-knee arteries using an Ascer arteriovenous fistula interposition is a good solution in patients with unavailable autologous conduits. This technique resulted in good long-term patency, limb salvage, and overall survival. Close duplex ultrasound follow-up is needed to prevent early graft thrombosis and to improve limb salvage.

## Figures and Tables

**Figure 1 diagnostics-13-01246-f001:**
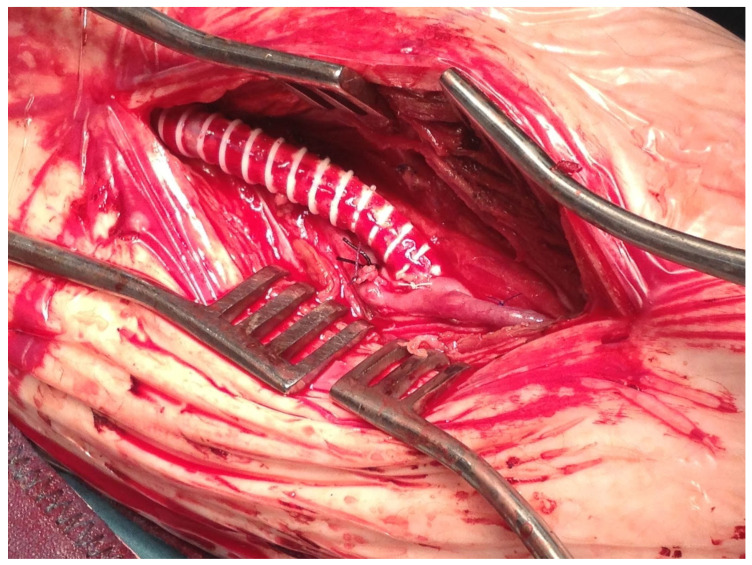
Distal arteriovenous fistula and PTFE anastomosis to the comitans vein.

**Figure 2 diagnostics-13-01246-f002:**
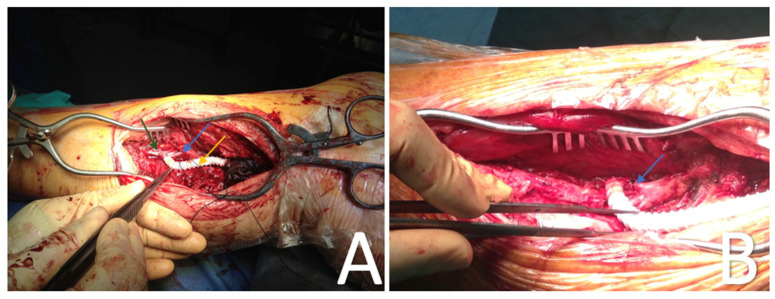
(**A**) Banding of the comitans vein with a silk suture. Green arrow: posterior tibial artery. Blue arrow: posterior tibial vein. Yellow arrow: PTFE graft. (**B**) Final result of AVF with a silk suture in the comitans vein.

**Figure 3 diagnostics-13-01246-f003:**
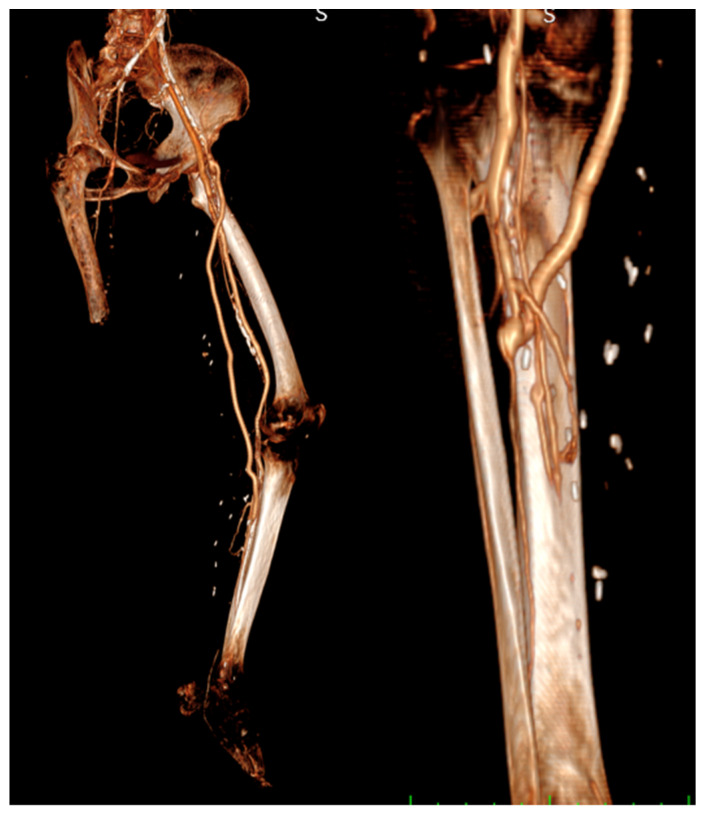
CT angiography.

**Figure 4 diagnostics-13-01246-f004:**
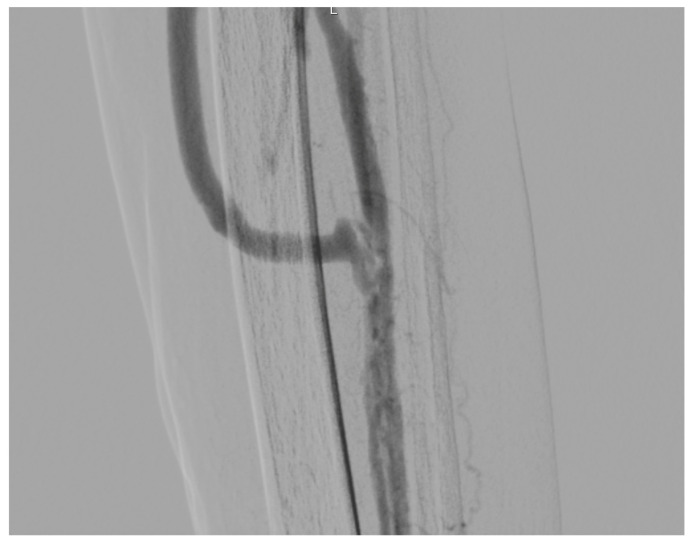
Angiography frame of the distal anastomosis on the peroneal artery.

**Figure 5 diagnostics-13-01246-f005:**
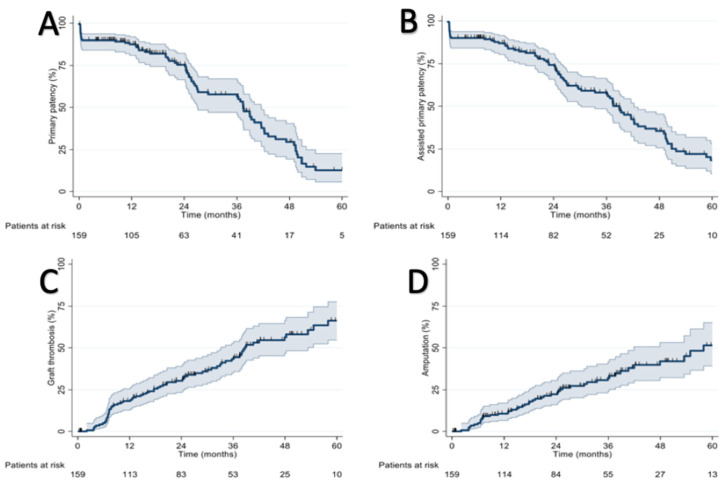
(**A**) Primary patency rates, (**B**) assisted primary patency rates, (**C**) graft thrombosis rates, and (**D**) amputation rates.

**Figure 6 diagnostics-13-01246-f006:**
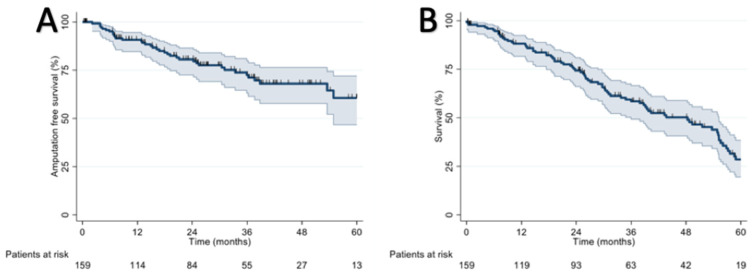
(**A**) Amputation-free survival rates and (**B**) survival rates.

**Table 1 diagnostics-13-01246-t001:** Demographic and baseline characteristics.

Age	73.8 ± 7.4
Sex	109 M (69%)
Smoking History	111 (80%)
Hypertension	98 (62%)
Dyslipidemia	137 (86%)
Previous Coronary Angioplasty	37 (23%)
Previous Coronary Bypass	39 (25%)
Diabetes	108 (68%)
Insulin-Dependent Diabetes	73 (46%)
Insulin-Independent Diabetes	35 (22%)
Chronic Obstructive Pulmonary Disease	81 (51%)
Chronic Kidney Disease (Creatinine > 1.3 mg/dL)	49 (31%)
Hemodialytic Treatment	21 (13%)
Rutherford 4	30 (19%)
Rutherford 5	72 (45%)
Rutherford 6	57 (36%)
WIfI 3	89 (56%)
WIfI 4	70 (44%)

**Table 2 diagnostics-13-01246-t002:** Technical bypass data.

Inflow Artery	
External Iliac Artery	27 (17%)
Common Femoral Artery	124 (78%)
Superficial Femoral Artery	8 (5%)
**Outflow Artery**	
Anterior Tibial Artery	45 (28%)
Posterior Tibial Artery	43 (27%)
Peroneal Artery	71 (45%)
**Comitans Vein for Distal Ascer Fistula**	
Anterior Tibial Vein	45 (28%)
Posterior Tibial Vein	49 (31%)
Peroneal Vein	65 (41%)
**PTFE graft Diameter**	
6 mm	89 (56%)
7 mm	64 (40%)
8 mm	6 (4%)

**Table 3 diagnostics-13-01246-t003:** Technical Data on Reintervention.

Total Reintervention	40 (25%)
PTA and Stenting on Inflow Vessels with Covered Stent	5 (3%)
PTA	35 (22%)
PTA of the Distal Anastomosis	15 (9%)
PTA on Outflow Vessels	20 (13%)
PTA with a DEB	16/20 (46%)

## Data Availability

Data are non available due to privacy.

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
