# Peer review of "Long-Term Results of Below-The-Knee Bypass Using a Prosthetic Graft with a Distal Arteriovenous Fistula Interposition"

_diagnostics, 2023, doi:10.3390/diagnostics13071246_

Round 1

Reviewer 1 Report

This is a report about unique bypass procedure. I have some questions.

1.     Please show the post-operative angiography around distal anastomosis to evaluate the arterial and venous flow. 

2.     Please describe the more details of the AVF procedures and it would be easier to understand in the schema were added to Figure 1. 

3.     What do you think about poor 5 years outcomes compared to 1 or 2 years?

Author Response

Response to Reviewer 1 Comments

Point 1: Please show the post-operative angiography around distal anastomosis to evaluate the arterial and venous flow. Response 1:  We provided an angiography frame of the distal anastomosis in the Materials and Methods section. 

Point 2: Please describe the more details of the AVF procedures and it would be easier to understand in the schema were added to Figure 1. 

Response 2: The AVF procedure is already described in Materials and Methods section as the author E. Ascer has already written. In figure 1, the prosthesis is anastomosed on the comitans vein and the distal banding with a silk suture is present.

 Point 3: What do you think about poor 5 years outcomes compared to 1 or 2 years?

Response 3: Patency results at five years are not satisfying and they are worse than the first two years. This is caused by the known progression of the atherosclerotic lesions, as reported in literature (Global Guidelines) the prosthesis have poor results in long-term patency also in bypass without distal adjunct in larger diameter’s arteries [1].

Reviewer 2 Report

This is an very interesting and well written paper that fits the Journal scope with a large population enrolled in the study. The aims of the presented study were to analyze the patency, limb salvage and survival rates of patients with CLTI, WIfI stage 3-4 and Glass stage 3 who undergoing to below-the-knee surgical revascularization using a prosthetic graft with a distal Ascer AVF fistula interposition.

            In materials and methods section, I suggest changing the Figure 1 and 2 with a higher resolution figure if the authors have that. Moreover, in Figure 3 I suggest introducing different angle of the same patient’s 3D CTA reconstruction with the distal anastomosis.

            In Results section, I suggest presenting all de information based of primary patency rates at 5 years, dividing all enrolled patients in two groups. Furthermore, the authors can make de multivariate analyzes to identify based on the abovementioned statistically significant difference between the 2 groups who are the predictors factors of primary patency failure and amputation rates.

             Overall, the manuscript is well written. Good luck with the revision.

Author Response

Response to Reviewer 2  Comments

Point 1: I suggest changing the Figure 1 and 2 with a higher resolution figure if the authors have that.

Response 1: Thank you for your question. We are so sorry, but these are the best pictures we have.

Point 2:Moreover, in Figure 3 I suggest introducing different angle of the same patient’s 3D CTA reconstruction with the distal anastomosis.

Response 2: Thank you for your question. We modified the Figure 3 and we added the posterior angulation that shows better the distal anastomosis.

Point 3: I suggest presenting all de information based of primary patency rates at 5 years, dividing all enrolled patients in two groups. Furthermore, the authors can make de multivariate analyzes to identify based on the abovementioned statistically significant difference between the 2 groups who are the predictors factors of primary patency failure and amputation rates.

Response 3: Thank you for this comment. In this first our data analysis we focused on the results at 5 years of the entire group we submitted to this kind of distal bypass. We think that it would be better creating 2 groups on anatomical or angiographic criteria or previous revascularization procedures. This could be a good proposal for the future.  

Round 2

Reviewer 2 Report

no further comments